# Continual few-shot learning with Hippocampal-inspired replay

## Abstract

Continual learning and few-shot learning are important frontiers in the quest to improve Machine Learning. There is a growing body of work in each frontier, but very little combining the two. Recently however, Antoniou et al. (2020) introduced a Continual Few-shot Learning framework, CFSL, that combines both. In this study, we extended CFSL to make it more comparable to standard continual learning experiments, where usually a much larger number of classes are presented. We also introduced an 'instance test' to classify very similar specific instances - a capability of animal cognition that is usually neglected in ML. We selected representative baseline models from the original CFSL work and compared to a model with Hippocampal-inspired replay, as the Hippocampus is considered to be vital to this type of learning in animals. As expected, learning more classes is more difficult than the original CFSL experiments, and interestingly, the way in which they are presented makes a difference to performance. Accuracy in the instance test is comparable to the classification tasks. The use of replay for consolidation improves performance substantially for both types of tasks, particularly the instance test.

## 1 Introduction

Over the past decade, Machine Learning (ML) has made impressive progress in many areas. The areas in which progress has been most dramatic share some common characteristics. Typically, a model learns from a large iid dataset with many samples per class and after a training phase, the weights are fixed i.e. it does not continue to learn. This is limiting for many applications and as a result distinct subfields have emerged which embrace different characteristics, such as continual learning and few-shot learning.

In continual learning (also known as lifelong learning), the challenge is to continually learn new tasks while maintaining performance on previous ones. A well known difficulty is catastrophic forgetting (McCloskey & Cohen, 1989) in which new learning disrupts existing knowledge. There are many approaches to tackle catastrophic forgetting that fall broadly into 3 categories (Delange et al., 2021): Regularization-based methods, Parameter isolation methods and Replay methods which are inspired by Hippocampal replay (Parisi et al., 2018). To our knowledge, none of the reported works explore continual learning with few samples per class.

In few-shot learning, only a few samples of each class are available. In the standard framework (Lake et al., 2015; Vinyals et al., 2017), background knowledge is first acquired in a pre-training phase with many classes. Then one or a few examples of a novel class are presented for learning, and the task is to identify this class in a test set (typically 5 or 20 samples of different classes). Knowledge of novel classes is not permanently integrated into the network, which precludes continual learning.

A special case of few-shot learning is reasoning about specific instances. This is easy for animals, but typically neglected by ML research. For example you usually know which coffee cup is yours, even if it appears similar to the cup of tea that belongs to your colleague. It is easy to see how this capability has applications across domains from autonomous robotics to dialogue with humans to fraud detection.

Another enviable characteristic of human and animal learning, is the ability to perform both continual and few-shot learning simultaneously. We need to accumulate knowledge quickly and may only ever receive a few examples to learn from. For example, given knowledge of vehicles (e.g. trucks, cars, bikes etc.), we can learn about any number of additional novel vehicles (e.g. motorbike, then skateboard) from only a few examples. This ability is critical for everyday life, particularly artificial agents in changing environments and many industry applications.

In order to combine continual and few-shot learning, Antoniou et al. (2020) introduced the Continual Few-Shot Learning (CFSL) framework. It is flexible, allowing the description of diverse scenarios with only a small set of parameters: the number of small training sets, referred to as 'support sets' (NSS), how many support sets before class changes (CCI), number of classes ($n$-way), and number of exposures ($k$-shot). The original study compared standard few-shot learning algorithms and the experiments were limited to only 5 classes per support set and a maximum of 10 support sets, whereas in continual learning problems, there are typically a much larger number of classes and support sets.

In this study we extended the work of (Antoniou et al., 2020) in three main directions. First, we ran experiments with an order of magnitude more classes to give results that are comparable with typical continual learning studies. Second, we introduced an instance test, which is a special case of the CFSL framework. Third, we investigated the effect of Hippocampal-inspired replay under these conditions.

## 2 Experimental Method

We first give an overview of the CFSL framework (Antoniou et al., 2020), upon which our study is based, in Section 2.1. Second, we describe experiments to scale selected tests in (Antoniou et al., 2020) to a greater number of classes, referred to as 'CFSL at scale', in Section 2.2. Third, we introduce the instance test in Section 2.3. Finally, we describe the models used in the experiments in Section 2.4. The source code for all experiments is located at `https://github.com/xxxx`.

### 2.1 Continual few-shot learning framework - background

As context, we will recap the CFSL framework and terminology. In continual learning, new tasks are introduced in a stream and old training samples are never shown again. Performance is continually assessed on new and old tasks. In the CFSL framework, the new data is presented with collections of samples defined as 'support sets', and then the model must classify a set of test samples in a 'target set'. The experiment is parameterised by a small set of parameters described in Table 1. By varying these parameters, the experimenter can control the total number of classes, NC, samples per class, and the manner in which they are presented to the learner. A visual representation is shown in Figure 1.

Table 1: The parameters that fully define an experiment in the CFSL framework by Antoniou et al. (2020)

| Parameter | Description |
| --- | --- |
| NSS | Number of support sets |
| CCI | Class-change interval e.g. if CCI=2, then the class will change every 2 support sets |
| $n$-way | Number classes per set |
| $k$-shot | Number of samples per support class in a support set |

### 2.2 CFSL at scale

We chose to base our experiments on the parameters of Task D as described in the original CFSL benchmark (Antoniou et al., 2020). Task D (see Figure 1) introduces both new classes and multiple instances of each class, which we argue is the most common, applicable real-world scenario.

We empirically optimized the experimental method for speed and efficiency. We observed the learning curves, and reduced the number of epochs and iterations where it did not sacrifice accuracy. In the case of one of

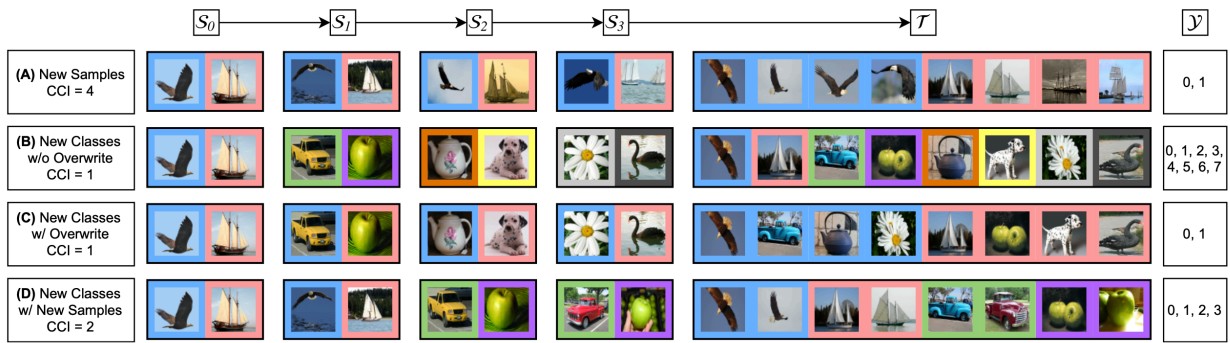

Visual representation of the four continual few-shot task types. Each row corresponds to a task with Number of Support Sets, NSS=4, and a defined Class-Change Cnterval (CCI). Given a sequence of support sets, $\mathcal{S}_n$, the aim is to correctly classify samples in the target set, $\mathcal{T}$. Colored frames correspond to the associated support set labels.

Figure 1: **Visual representation of CFSL experiment parameterisation.** Reproduced from (Antoniou et al., 2020).

the models, VGG (Simonyan & Zisserman, 2014), that meant lowering from 250 epochs and 500 iterations per epoch to 10 epochs and 100 iterations per epoch. In addition, we decreased the number of tasks from 600 to 100, without noticeably affecting the mean and standard deviation accuracy across tasks.

In the original CFSL benchmark, Antoniou et al. (2020) took an ensemble of models from the top 5 performing iterations. Ensembling is well-known to improve performance in ML, and so we preferred to see a more direct measure of performance by simply taking the mean across tasks. We include the ensemble scores for comparison. The original experiments were conducted on both Omniglot and Slimagenet datasets. Our study utilizes Omniglot as a starting point.

### 2.2.1 Replication

During our work with CFSL, we identified and fixed a number of issues in the original CFSL codebase (Antoniou et al., 2020), and collaborated with the authors to have them reviewed and merged upstream. Given the significance of some of these issues, we opted to replicate a selected number of the original experiments to properly contextualise our new experiments and results. The main issues related to a) VGG weight updates and b) mislabelling of new instances which became an issue where CCI>1.

### 2.2.2 Scaling

One of the limitations in (Antoniou et al., 2020) is the small total number of classes in each experiment (5 classes per support set and a maximum of 10 support sets). It is common in the continual learning field, for the number of classes to range from 20 to 200, even if the number of tasks in a sequence may be small (approximately 10). Therefore, we introduced experiments with up to 200 classes, presented in two ways. **Wide**, in which the number of support sets was small but with a larger number of classes per set, and **Deep**, where there were a larger number of support sets but with a smaller number of classes per set. See Figure 2.

Two of the replication configurations were used as baselines, with 10 and 20 classes. Then, we created Wide and Deep configurations, with 20 total classes like the 2nd baseline, but we modified the way that the classes were presented. Finally, the number of classes was increased ten-fold to 200, presented in both Wide and Deep configurations. In all of the experiments, $k$-shot is set to 1, so for any support set, there is only one exemplar per class.

To ensure that the scaling experiments are a fair comparison to the replication experiments, particularly given that the experiment size is dramatically increased, we conducted extensive hyperparameter search to empirically optimize the results.

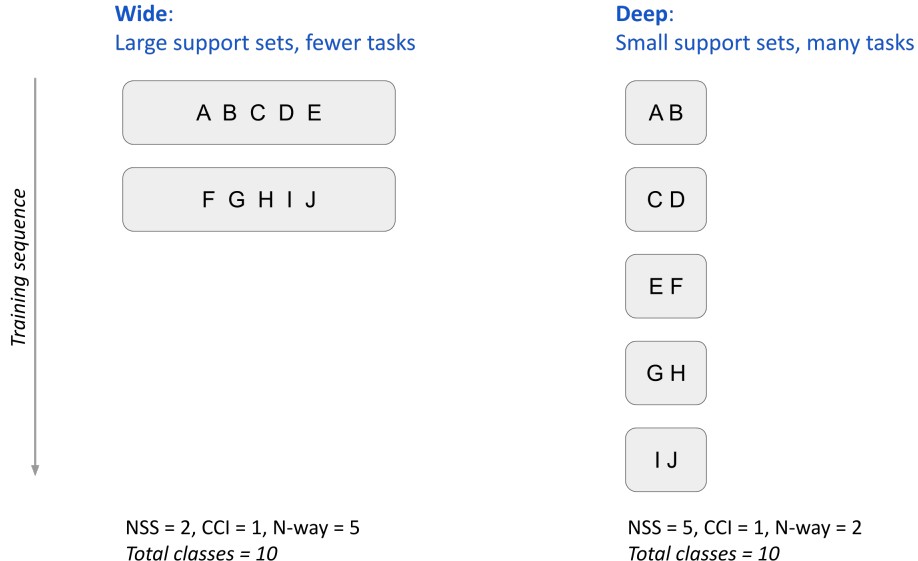

Figure 2: **Wide vs Deep.** An illustration of Wide vs Deep experiments. Wide have big support sets and few tasks, Deep has small support sets and many tasks.

## 2.3 Instance test

The standard definition of 'classification' concerns generalization. In other words, the objective is to learn to ignore small variations that are not statistically significant. In contrast, learning specific instances requires an ability to learn the distinct characteristics of a specific instance of a class, to differentiate between very similar samples, even to differentiate samples of the same class. It implies memorization, but still requires a degree of generalization i.e. you need to be able to recognize the same instance despite observational variation caused by factors such as lighting changes or occlusion.

Reasoning about instances is also crucial for intelligent agents, and is something that we take for granted in humans and other animals. For example, knowing your own mug from others' cups or mugs, in addition to recognising that it belongs to the 'cup' class. More generally, it underpins memory for singular facts and an individual's own autobiographical history, important for future decision making.

To evaluate the capabilities described above, we invented an 'instance test' version of CFSL based on the instance test in AHA (Kowadlo et al., 2020). In our instance test, the learner must learn to recognize specific exemplars amongst sets where all the exemplars are drawn from the same class.

It is a demonstration of the flexibility of the CFSL framework, that the instance test can be implemented as a special case of the existing parameters. $n$-way is set to 1, so that there is only one class in each support set. CCI is equal to NSS so that there is no class change between support sets. Then the $k$-shot or samples per class, determines how many instances are shown for a given class, which we refer to as Number of Instances, NI. We used a constant total number of instances for all experiments, 20, but experimented with presenting the samples differently, in terms of number and size of support sets. We reused empirically optimal hyperparameters from the Scaling Experiments.

## 2.4 Models

### 2.4.1 Baseline architectures

We selected three models from the original CFSL paper to use as baselines, to represent each family of algorithm that was tested. The first model was VGG (Simonyan & Zisserman, 2014), which is a standard

CNN architecture, trained with conventional minibatch stochastic gradient descent. The second model was ProtoNet (Snell et al., 2017), which is a meta-learning approach. We intended to also evaluate SCA, which is a complex and high performing meta-learning approach. Unfortunately however, we encountered scalability and resource issues with SCA. We were unable to successfully complete the larger variant of each experiment type for the SCA model. The experiments for the VGG baseline and ProtoNets were achievable with our available computational resources.

### 2.4.2 Learning with replay

A core objective of this research is to test the concept of CLS (McClelland et al., 1995; O'Reilly et al., 2014; Kumaran et al., 2016) replay in CFSL. In CLS, a short term memory (STM) stores recent representations of stimuli in a highly non-interfering manner. Interleaved replay to a long term memory (LTM), which is assumed to be an iterative statistical learner, enables improved learning and retention of that knowledge.

We utilise a simple circular buffer as STM. It is an idealised version of a Hippocampal STM, and as such sets an upper bound on replay benefit, which serves as a useful first test of the concept. The buffer is idealised in the sense that it provides perfect memorization and recall of input. It would be interesting to use a biologically plausible STM such as AHA (Kowadlo et al., 2019; 2020; 2021), which we leave for future work.

A disadvantage of the buffer implementation is the use of a naive nearest neighbour lookup for recall, whereas a more biologically realistic algorithm such as AHA would be better able to generalise i.e. recall a previous exemplar from a novel exemplar presented as a cue. Another disadvantage is that the buffer stores the full input images. An algorithm such as AHA has the potential to store more abstract embeddings, with additional memory efficiency benefits.

The VGG baseline is the most straightforward and appropriate model to pair with a short term memory in a CLS architecture. The architecture is shown in Figure 3. In the original CFSL task without a replay buffer, the VGG is pre-trained on a background set of images. During the CFSL few-shot phase, when a support set is presented during training, it is used to fine-tune the VGG. When using replay, there are two stages. First, the current support set is stored in the STM, adding to recent support sets. $b$ is the buffer size, measured in support sets, and is a tuneable hyperparameter. Second, the VGG is trained using the current support set, as well as samples randomly drawn from the replay buffer. The number of samples is determined by a second hyperparameter $k$. We conducted a hyperparameter search to optimise for $b$ and $k$ across the different experiments.

Adding the replay buffer increased the memory requirements. For some experiments, we reduced the number of fine-tuning training steps to make it possible to run within our hardware constraints. Replay requires more than 1 support set, and so it is not relevant for one of the configurations of the instance tests, Experiment 1, which contains only 1 support set.

## 3 Results

### 3.1 CFSL at scale

#### 3.1.1 Replication

The results of the replication experiments are summarised in Table 2, which includes reference values from (Antoniou et al., 2020) for comparison. In the experiments that were affected by code fixes (all VGG runs and ProtoNets where CCI>1), performance improved substantially from unusually low values, and the performance across experiments followed a more expected trend (i.e. increasing accuracy of VGG with decreasing number of classes). ProtoNets are substantially more accurate than VGG, and perform consistently across different variations of the presentation of 10-50 total classes.

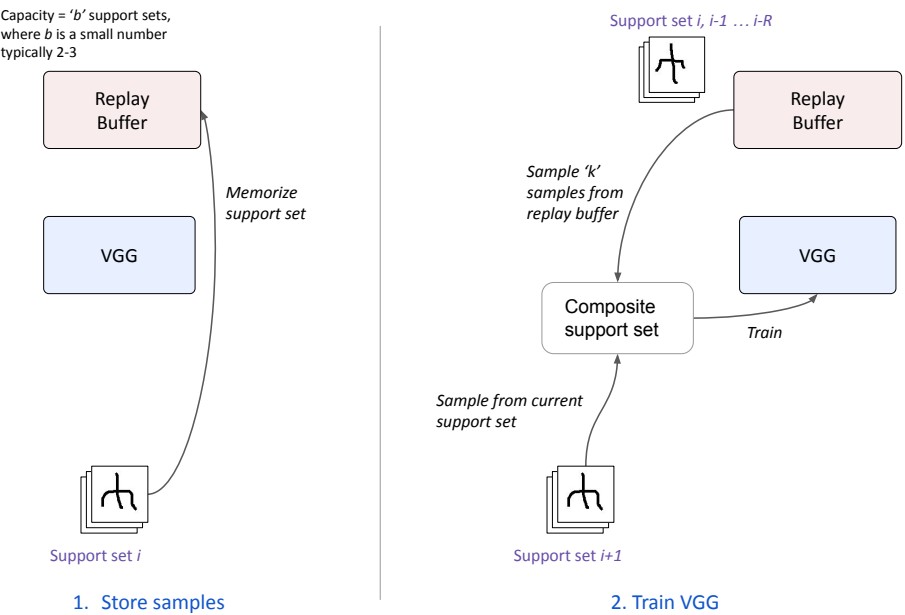

Figure 3: **Learning with replay.** CLS setup with VGG Long Term Memory (LTM), paired with a circular buffer Short Term Memory (STM). First, in a memorization step, the STM temporarily stores recent support sets. Second, in a recall step, the memorizaed data are used in LTM training.

Table 2: **Replication experiments.** Replication of Task D from (Antoniou et al., 2020) after correcting errors in the framework code. Accuracy is shown as mean ± standard deviation across 3 random seeds. All accuracies are in %.

| Model name | NSS | CCI | $n$-way | $k$-shot | Number of classes | Ensemble Accuracy | Accuracy | Reference Ensemble Accuracy |
|---|---|---|---|---|---|---|---|---|
| **VGG** | 4 | 2 | 5 | 2 | 10 | $37.81 \pm 0.77$ | $36.7 \pm 0.80$ | $7.91 \pm 0.15$ |
| **VGG** | 8 | 2 | 5 | 2 | 20 | $27.92 \pm 0.10$ | $26.41 \pm 0.14$ | $3.86 \pm 0.06$ |
| **VGG** | 3 | 1 | 5 | 2 | 15 | $17.76 \pm 0.32$ | $17.40 \pm 0.33$ | $9.97 \pm 0.14$ |
| **VGG** | 5 | 1 | 5 | 2 | 25 | $13.76 \pm 0.08$ | $13.10 \pm 0.03$ | $6.02 \pm 0.02$ |
| **VGG** | 10 | 1 | 5 | 2 | 50 | $9.73 \pm 0.06$ | $8.36 \pm 0.05$ | $3.13 \pm 0.03$ |
| **ProtoNets** | 4 | 2 | 5 | 2 | 10 | $97.93 \pm 0.05$ | $96.98 \pm 0.05$ | $48.98 \pm 0.03$ |
| **ProtoNets** | 8 | 2 | 5 | 2 | 20 | $96.66 \pm 0.03$ | $95.22 \pm 0.06$ | $48.44 \pm 0.03$ |
| **ProtoNets** | 3 | 1 | 5 | 2 | 15 | $97.12 \pm 0.06$ | $95.88 \pm 0.12$ | $95.30 \pm 0.12$ |
| **ProtoNets** | 5 | 1 | 5 | 2 | 25 | $95.93 \pm 0.12$ | $94.36 \pm 0.05$ | $91.52 \pm 0.20$ |
| **ProtoNets** | 10 | 1 | 5 | 2 | 50 | $92.43 \pm 0.27$ | $90.24 \pm 0.10$ | $83.72 \pm 0.19$ |

### 3.1.2 Scaling

The results are summarised in Table 3 and Figure 4. The best set of hyperparameters are shown in Appendix 8.1. The number of fine-tuning training steps for the replay experiments, which had to be reduced to allow it to run within our hardware RAM constraints, are shown in Appendix 8.2.

Increasing the number of classes by an order of magnitude (to 200) led to a dramatic decrease in accuracy (to only approximately 5%). The manner in which the classes were presented makes a difference to learning. Firstly, rearranging the presentation of 20 classes from Baseline 2 to Wide or Deep improved performance.

Table 3: **Scaling test.** The table shows experiments (columns) for each model (rows). These results are for the best configurations found through hyperparameter search. Accuracy is shown in %, as mean ± standard deviation across 5 random seeds. NC=number of classes.

| Model name | Baseline 1 (NSS=4, CCI=2, $n$-way=5, NC=10) | Baseline 2 (NSS=8, CCI=2, $n$-way=5, NC=20) | Wide 1 (NSS=4, CCI=2, $n$-way=10, NC=20) | Wide 2 (NSS=4, CCI=2, $n$-way=100, NC=200) | Deep 1 (NSS=20, CCI=2, $n$-way=2, NC=20) | Deep 2 (NSS=80, CCI=2, $n$-way=5, NC=200) |
|---|---|---|---|---|---|---|
| **VGG** | $64.95 \pm 1.00$ | $33.71 \pm 3.54$ | $54.25 \pm 0.59$ | $4.22 \pm 0.50$ | $33.44 \pm 1.18$ | $6.64 \pm 0.68$ |
| **ProtoNets** | $86.67 \pm 1.35$ | $88.04 \pm 1.12$ | $86.92 \pm 0.42$ | $65.61 \pm 9.31$ | $88.56 \pm 0.61$ | $80.30 \pm 1.15$ |
| **VGG+replay** | $81.15 \pm 0.81$ | $73.5 \pm 0.43$ | $80.74 \pm 0.70$ | $31.78 \pm 0.65$ | $60.94 \pm 0.90$ | $18.62 \pm 0.71$ |

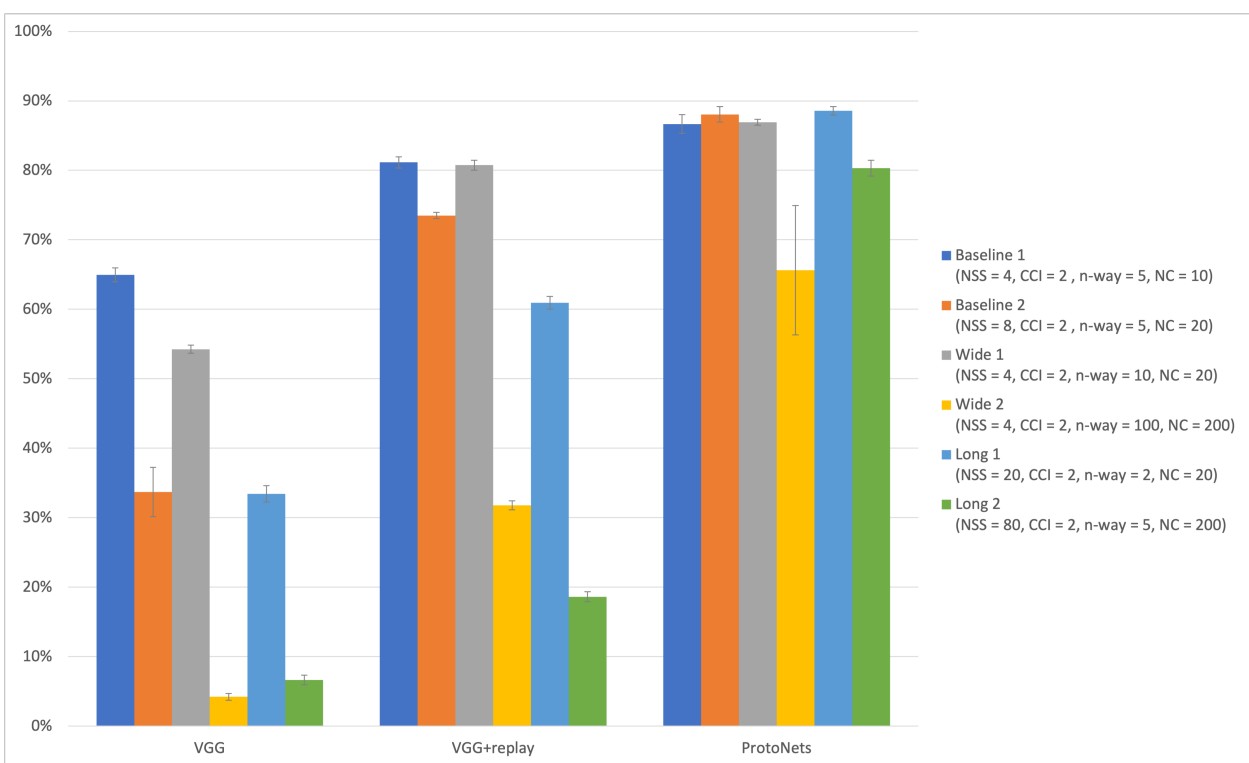

Figure 4: **Scaling experiments.** These results are for the best configurations found through hyperparameter search. Error bars show 1 standard deviation across 5 random seeds.

Secondly, the decreased performance from Baselines 1 and 2 to Wide and Deep is more pronounced for the Deep experiment.

ProtoNets showed a similar profile with a few exceptions. The overall accuracy is much better than VGG. Doubling the number of classes between Baseline 1 and 2 did not decrease performance. In fact it improved slightly (although the means are within a standard deviation). Decrease in performance with 200 classes is large but modest in comparison to VGG, approximately 23% and 10% proportional decrease for Wide and Deep respectively.

For both VGG and ProtoNets, the Wide experiments were more difficult compared to Deep. For VGG, replay substantially improved performance, ranging from approximately 12% to 40%, without a clear trend over the experimental configurations.

### 3.2 Instance test

The results for the instance test are summarised in Table 4 and Figure 5. We identified the most comparable configuration from the scaling experiments as Baseline 2, and used the same hyperparameters (see Appendix 8.1). The number of fine-tuning training steps for the replay experiments, which had to be reduced to allow it to run within our hardware RAM constraints, are shown in Appendix 8.2.

The number of 'items to identify', which in this case are separate instances, is constant at 20 for all of the experiments. For VGG, the performance is variable depending on how the instances are presented. It is in the same range as the scaling experiments with the same number of 'items to identify', which in that case, are total classes. Accuracy increases as the instances are distributed over more, smaller support sets. The trend of increasing accuracy with increased support sets starts to reverse when the number of support sets becomes too large. However, this is likely due to the fact that the subsequent decrease in support set size becomes pathological at only one.

Replay boosts performance substantially, from approximately 40% to 96% for Experiment 2. The improvement remains substantial but decreases as the number of support sets increases, and their size decreases.

ProtoNets are again effective and insensitive to experiment configuration, even more so than for the scaling experiments. For VGG, replay substantially improved performance, ranging from 12% to 40%, beating ProtoNets in some cases.

Table 4: **Instance test.** These results are for the best configurations found through hyperparameter search. Accuracy is shown in %, as mean ± standard deviation across 5 random seeds. $n$-way=1 for all experiments, to restrict distinguishing between similar instances of a single class. In the instance test, $k$-shot translates to the size of the support set. It is 1-shot in the sense that each instance is only shown once. NI, number of instances = 20 for all the experiments.

| Model name | Exp.          1 (NSS=1, $k$-shot=20, NI=20) | Exp.          2 (NSS=2, $k$-shot=10, NI=20) | Exp.          3 (NSS=4, $k$-shot=5, NI=20) | Exp.          4 (NSS=10, $k$-shot=2, NI=20) | Exp.          5 (NSS=20, $k$-shot=1, NI=20) |
|---|---|---|---|---|---|
| **VGG** | $18.33 \pm 0.76$ | $40.35 \pm 1.40$ | $47.63 \pm 5.00$ | $47.72 \pm 2.77$ | $43.88 \pm 2.47$ |
| **ProtoNets** | $92.72 \pm 1.01$ | $92.38 \pm 1.00$ | $91.94 \pm 1.09$ | $91.79 \pm 1.16$ | $91.79 \pm 1.16$ |
| **VGG+replay** | N/A | $96.39 \pm 1.36$ | $89.19 \pm 2.79$ | $82.77 \pm 2.15$ | $79.46 \pm 3.01$ |

## 4 Discussion

In this study we found that for the architectures tested, few-shot continual learning of new classes is more difficult at scale (the 'scaling test') i.e. as the number of classes increased from 20 to 200. A novel 'instance test' was approximately as difficult for each model as similar sized classification tasks in the 'scaling test'. ProtoNets outperformed VGG in all tasks (scaling test and instance test), but with the addition of replay, VGG+replay accuracy improved substantially becoming comparable to ProtoNets on the instance test.

### 4.1 Model comparisons

The experiments involved two model types: VGG and ProtoNets (Snell et al., 2017). It's natural to compare their performance, but comparison should be cautious as ProtoNets and VGGs do not perform the same learning task.

VGG (Simonyan & Zisserman, 2014) is a CNN architecture, and is trained for classification. On the other hand, ProtoNets (Snell et al., 2017) learn embeddings that can be *used* for classification. Classification is achieved by comparing embeddings, which necessitates a short-term memory (STM) of the reference embedding being matched. By convention, that memory is in the testing framework rather than the ProtoNet architecture itself. From this perspective, adding the STM (replay buffer) to VGG makes it similar to

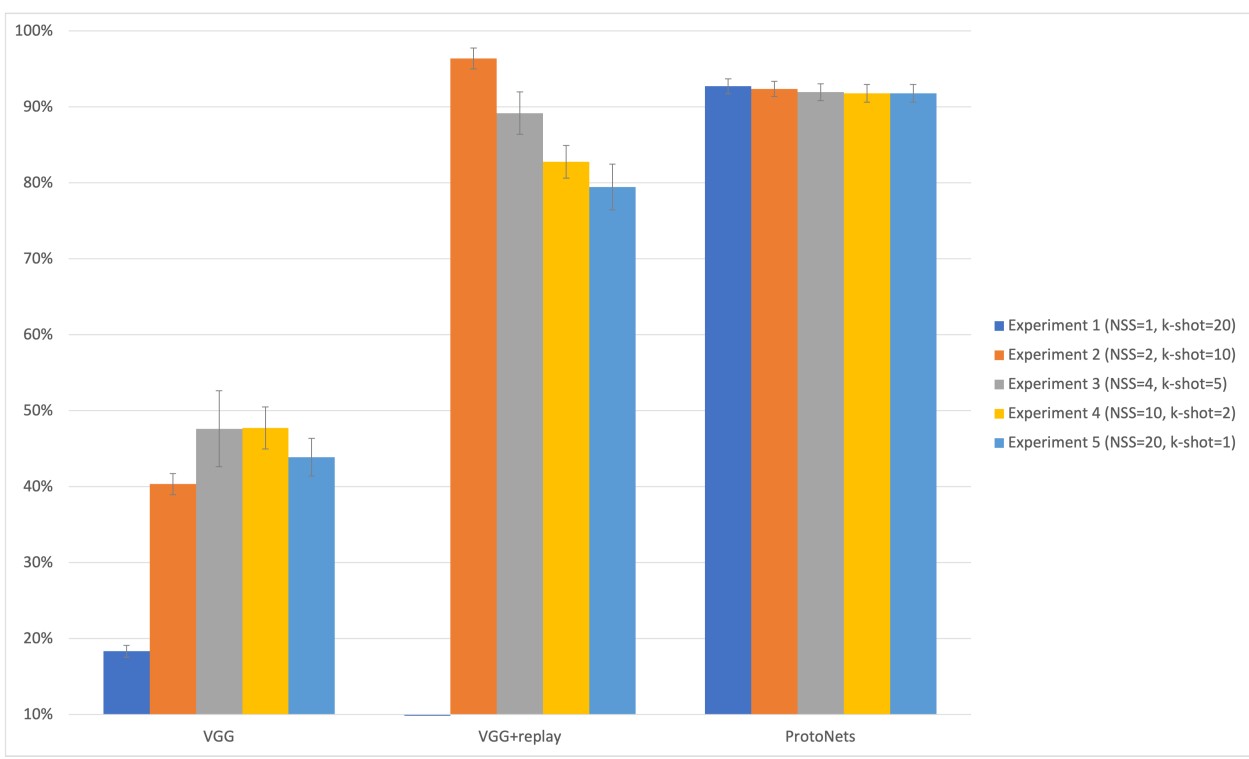

Figure 5: **Instance test.** These results are for the best configurations found through hyperparameter search. Error bars show 1 standard deviation across 5 random seeds. In the instance test, $k$-shot translates to the size of the support set. It is 1-shot in the sense that each instance is only shown once. NI, number of instances = 20 for all the experiments.

ProtoNets. In this study the STM is used for replay only. In future work, it will also be used for classification for recent samples still in short term memory, as was done in AHA (Kowadlo et al., 2021). In the case of ProtoNets, the objective of representation learning is to optimize one-shot class generalization. The objective of AHA is to learn representations that are *also* highly differentiated, which in theory should be better at the instance test.

Another way in which VGG and ProtoNet learning differs, is that the ProtoNet architecture in (Antoniou et al., 2020) and our study does not actually acquire new knowledge during training and therefore does not continually learn. There is just one optimizer (for the meta-learning 'outer loop') that gets triggered during the 'pre-training' phase, meaning that it learns meta parameters for an embedding representation that will be optimal for downstream tasks. Since these ProtoNets do not learn, they also do not forget older knowledge and VGG+replay may prove more effective when consolidating new knowledge compared to ProtoNets' fine-tuning on additional training data (explored further below in the context of replay, Section 4.4). We note that in the original ProtoNets paper (Snell et al., 2017), there is fine-tuning of weights after the initial pre-training.

## 4.2   Wide vs Deep - big batches vs many tasks (no replay)

The way that data are presented, not just the number classes, made a difference to learning. When the number of classes was held constant at 20, but the number and size of support sets was varied (Baseline 2 vs Wide 1 or Deep 1), performance was better for the wide configuration. However, when the number of classes was increased by an order of magnitude to 200 (Wide 2 vs Deep 2), the performance was better in the deep configuration, where the classes were spread out across smaller batches. This was unexpected given that weight updates (in VGG) occur after each support set, and the more support sets there are, the more it could 'forget' earlier learning. It is possible that as the number of classes increase, the larger support sets (in the Wide experiments) are harder to learn, or cause sharper forgetting by virtue of the fact that more knowledge is being acquired in one update.

ProtoNets (Snell et al., 2017) are very effective in the scaling test, despite not actually learning during these tasks. The results show that they learnt an effective embedding space during pre-training for the task. Since no learning takes place, performance cannot suffer due to forgetting. Therefore lower performance when there are a lot of classes, as in Wide 2 and Deep 2, is likely due to generation of similar embeddings for different classes.

## 4.3   Specific instances (no replay)

General performance on the instance test was surprisingly good. Compared to classification in experiment configurations with a comparable 'number of items', ProtoNets were more accurate and stable. VGG+replay accuracy was in a similar range, in some cases better and others worse. The results suggest that one-shot distinguishing of specific (very similar) instances is not more difficult than classification, for these architectures.

The fact that VGG accuracy increases as the instances are distributed over more support sets, further hints that CNNs may be more effective at continual learning with smaller support sets, which is in-line with our interpretation of why Deep (more, smaller support sets) was easier than Wide in the scaling experiments (explained in more detail in Section 4.2 above).

ProtoNets are effective in the instance test as well as classification. In the instance test, generalization is not required, and so the representations are less likely to overlap reducing the possibility of clashes. In addition, no fine-tuning occurs (see earlier in the Discussion, Section 4.1), so performance is very stable across all configurations. If noise or occlusion were introduced, a degree of generalization would be required, and it is likely that the performance would suffer. These conditions were explored in (Kowadlo et al., 2020).

### 4.4 Effect of replay

As hypothesized, adding replay to VGG enabled a strong improvement across tasks. Despite the improvement, performance did not reach the same level as ProtoNets in all scaling tests (classification). Replay had a more substantial impact in the instance test, compared to the scaling test. VGG+replay outperformed ProtoNets where NSS=2, the accuracy was within one standard deviation for NSS=4, and gradually fell lower than ProtoNets for NSS=10 or 20. However, this deterioration may be due to the fact that we reduced the number of fine-tuning iterations to allow it to run within our resource constraints, specifically GPU RAM (see Appendix 8.2).

Why is replay more effective in the instance test? The number of samples available for replay is limited to only 2 support sets, making it prone to overfitting, which likely provided an advantage in the instance test.

Although VGG+replay did not convincingly allow VGG to outperform ProtoNets on these experiments, the enhanced performance has a number of implications for future work. Replay improves performance of a statistical learner (i.e. an LTM) in few-shot continual learning and the VGG long term memory (LTM) could easily be replaced with other more sophisticated models such as ResNet (He et al., 2015). Moreover it is possible that the relative advantage of ProtoNets will decrease for more complex datasets, which did occur for several models in the original CFSL experiments (Antoniou et al., 2020). Under these conditions an effective LTM with replay could be more effective than ProtoNets or other alternatives.

Finally, and perhaps most significantly, ProtoNets as implemented do not acquire new knowledge during training, as explained earlier in this section, and therefore do not actually demonstrate continual learning. The inability to adapt is very likely to limit performance if there is a shift in the statistics of data distribution. Pushing these limits and exploring weight adaptation during training in the context of CFSL is an important area for future work.

## 5 Limitations

The base framework measures performance after all the learning has occurred. In contrast, most studies in the continual learning literature document progressive performance as new tasks are introduced, which gives more visibility into learning. Also, the models and tasks used are very computationally expensive, limiting our ability to compare to some of the models used in the original work.

## 6 Future Work

A promising direction for future work is to use or develop a more sophisticated Hippocampal model to improve CFSL. For example, like in AHA (Kowadlo et al., 2019; 2021), an ability to store compressed representations for more efficient storage, use of the STM (together with the LTM) for inference, not just consolidation, and multiple pathways to support both pattern separation (instance test) and generalization (classification), and like in (Lu et al., 2022), an ability to selectively encode and retrieve memories for more efficient storage and training.

## 7 Conclusion

In this study we scaled the CFSL framework to make it more comparable to typical continual learning experiments. We introduced two variants, **Wide** with fewer larger training 'support sets' and **Deep** with a greater number of smaller support sets. We also introduced a few-shot continual instance test, which is important in everyday life, but often neglected in Machine Learning. Increasing the number of classes decreased classification performance (scaling test) and the way that the data were presented did make a difference to accuracy. Performance in the few-shot instance test was comparable to few-shot classification. Augmenting VGG with a replay algorithm improved performance substantially, especially in the instance test. In most experiments, replay did not boost performance above ProtoNets, but the utility of replay is clearly demonstrated, and an LTM with replay architecture may be more effective than ProtoNets or other models under different experimental conditions.

Continual few-shot learning, for both classes and instances, is a necessary capability for agents operating in unfamiliar and changing environments as well as providing new possibilities for business applications where these conditions may often occur. This study is one of the first steps in that direction, combining continual and few-shot learning, and it demonstrates that Hippocampal-inspired replay is a promising approach.

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

## 8 Appendix

### 8.1 Hyperparameters for VGG+replay

Table 5: **Hyperparameters.** Optimized Hyperparameters for the scaling test and VGG+replay. lr is the learning rate of the Long Term Memory, $b$ is the replay buffer size measured in number of support sets, and $k$ is the number of samples randomly drawn from the replay buffer for training the Long Term Memory.

| Experiment | Hyperparameters |
|---|---|
| Baseline 1 | 512 filters, 3 stages, lr=0.01, $b$=2, $k$=10 |
| Baseline 2 | 128 filters, 3 stages, lr=0.01, $b$=4, $k$=10 |
| Wide 1 | 256 filters, 3 stages, lr=0.01, $b$=2, $k$=20 |
| Wide 2 | 128 filters, 2 stages, lr=0.01, $b$=2, $k$=50 |
| Deep 1 | 256 filters, 3 stages, lr=0.01, $b$=5, $k$=10 |
| Deep 2 | 256 filters, 3 stages, lr=0.01, $b$=5, $k$=10 |

### 8.2 Fine-tuning steps

Table 6: **Fine-tuning for scaling test.** The number of fine-tuning training steps for the VGG+replay scaling test.

| Experiment | Fine-tuning training steps |
|---|---|
| Baseline 1 | 120 |
| Baseline 2 | 60 |
| Wide 1 | 30 |
| Wide 2 | 30 |
| Deep 1 | 5 |
| Deep 2 | 5 |

Table 7: **Fine-tuning for the instance test.** The number of fine-tuning training steps for the VGG+replay instance test.

| Experiment | Fine-tuning training steps |
|---|---|
| Exp. 1 | N/A |
| Exp. 2 | 120 |
| Exp. 3 | 120 |
| Exp. 4 | 60 |
| Exp. 5 | 30 |

