# OpenReview forum: "Continual few-shot learning with Hippocampal-inspired replay"
_TMLR — Withdrawn by Authors_

### Review · Reviewer_SGAe · 2022-10-07

**Summary Of Contributions:**

The paper investigates the Continual Few-Shot Learning (CFSL) setting by building on top of the work of Antoniou et al. (2020). An element of novelty is the use of a memory replay mechanism roughly based on the biological counterpart. The authors report empirical results on the Omniglot dataset, introducing a few interesting conditions such as wide/deep sequences of tasks, and instance-based testing. The results show that the use of a memory replay is beneficial in many conditions. Overall the paper has potential, it investigates CFSL, which is an important setting that did not receive enough attention. However, I have major concerns regarding the empirical evaluations and some technical choices. I have detailed my doubts below. I think that the paper could be improved significantly but in its current form is not ready for publication.

**Broader Impact Concerns:**

No concern.

**Requested Changes:**

Requested Changes
----------------

1. The paper does not take into account memory usage and multiply-accumulate operations (MACs), this was one of the main points of Antoniou et al. (2020). This is particularly relevant here, since the use of a replay buffer can increase memory usage and overall MACs. I expect a discussion (possibly supported by data) on how the buffer replay affects memory and MACs.
2. Experimental results are based on suboptimal/obsolete backbones (VGG). It would be better to use more recent architectures, such as ResNets and EfficientNets variants. In particular, EfficientNets have been recently used in few-shot learning with state-of-the-art results (Bronskill et al., 2021); they are GPU-friendly as they have a low memory footprint (much lower than VGG) and could be easily tested on the proposed benchmark.
3. Results are based on a simple dataset (Omniglot). The few-shot learning community has moved towards more challenging datasets, such as MetaDataset (Triantafillou et al. 2019). For more realistic experiments, it is necessary to provide results on more complex dataset. Scaling to the SlimageNet dataset of Antoniou et al. (2020) could be the easier solution here.
4. Replay memory. This mechanism needs to be better described in the paper since it has been claimed as one of the main contributions.
5. I have some doubts regarding the effectiveness of the replay memory mechanism. The authors mention that the replay memory uses a naive nearest neighbour lookup for recall (Section 2.4.2). This could work relatively well on simple datasets like Omniglot, but it may not work on more complex datasets (e.g. ImageNet-like dataset). This further supports the need of repeating the experiments on more complex datasets.
6. There is no discussion of related work. A section should be included discussing previous work. Some relevant papers are Ren et al. (2020), Caccia et al. (2020), and Harrison et al. (2020). In particular the discussion should focus on how the work can be framed and contextualized after the upgrades made to the paradigm of Antoniou et al. (2020).
7. It is not clear how the hyper-parameters have been found. Did the authors use a validation set for this purpose? Searching the hyper-parameters on the training set could be inappropriate and lead to biased results.
8. Regarding the instance test (Section 2.3) it is not clear to me how those instances have been built. It would be helpful if the authors provide some graphical examples (e.g. in the appendix).
9. I would rename the method that the author called VGG, as this is the name of the underlying backbone and not the method of the technique that has been used.
10. The authors mention "hardware constrains" various times in the paper but I did not find a description of the hardware used. Details about the hardware and the duration of the training phase must be reported.

References
----------

Antoniou, A., Patacchiola, M., Ochal, M., & Storkey, A. (2020). Defining benchmarks for continual few-shot learning. arXiv preprint arXiv:2004.11967.

Bronskill, J., Massiceti, D., Patacchiola, M., Hofmann, K., Nowozin, S., & Turner, R. (2021). Memory efficient meta-learning with large images. Advances in Neural Information Processing Systems, 34, 24327-24339.

Caccia, M., Rodriguez, P., Ostapenko, O., Normandin, F., Lin, M., Caccia, L., ... & Charlin, L. (2020). Online fast adaptation and knowledge accumulation: a new approach to continual learning. arXiv preprint arXiv:2003.05856.

Harrison, J., Sharma, A., Finn, C., & Pavone, M. (2020). Continuous meta-learning without tasks. Advances in neural information processing systems, 33, 17571-17581.

Ren, M., Iuzzolino, M. L., Mozer, M. C., & Zemel, R. S. (2020). Wandering within a world: Online contextualized few-shot learning. arXiv preprint arXiv:2007.04546.

Triantafillou, E., Zhu, T., Dumoulin, V., Lamblin, P., Evci, U., Xu, K., ... & Larochelle, H. (2019). Meta-dataset: A dataset of datasets for learning to learn from few examples. arXiv preprint arXiv:1903.03096.

**Strengths And Weaknesses:**

Strengths
----------

- The authors focus on a very important topic at the intersection of Continual and Few-shot learning.
- Some of the results are interesting, for instance the fact that the final performance is affected by the way classes are presented.
- The instance test, introduced in Section 2.3, is an interesting addition to the evaluation pipeline.

Weaknesses
----------------

- The experiments are based on obsolete architectures and simple datasets.
- The description and motivation of the Memory Replay mechanism is not exhaustive.
- Comparison with relevant previous work is missing.

---

### Review · Reviewer_TSHH · 2022-10-13

**Summary Of Contributions:**

This paper studies continual few-shot learning following the setup presented in previous work. The main contribution is to explore two settings that weren’t included in the previous study: larger number of classes being learned, and an instance classification variant where each example can be seen as its own ‘class’. They experimentally compare ProtoNets to a VGG classifier and experiment adding replay to the latter. All experiments are performed in Omniglot. The main conclusions from the experiments are the following. ProtoNets outperform VGG in almost all cases. Adding replay helped VGG, sometimes significantly (especially for the instance test), but it almost always still lags behind ProtoNets. Increasing the number of classes significantly makes the problem harder, unsurprisingly, and the way in which the classes are presented (many classes in a small number of support sets, versus fewer classes sequentially in a larger number of support sets) affects performance.

**Broader Impact Concerns:**

No concerns.

**Requested Changes:**

Clarity
======
There are a few key points where clarity is poor. Firstly, it should be clearly explained how each model is (pre-)trained and how exactly it is used to tackle each task. I am inferring from discussion of experimental results that VGG is simply trained as a classifier (as usual) and then is finetuned in each few-shot task (but unclear what the output head is; I’m assuming a new one is initialized randomly for each episode? I’m assuming the model parameters are finetuned too in the process?). For ProtoNets, what architecture was used? This is a factor that may influence the results greatly (perhaps some of the differences observed are due to different capacity for the two models compared). This is all very important and needs to be described for each model in the section describing the models, before the experiments are presented.

Secondly, when presenting their ‘instance test’, the authors clearly describe how the support sets are formed, but an important omission is the description of the target set. It is not clear how ‘held-out’ examples of a specific *instance* can be used to form the target set. One idea would be to use data augmentation, as is done in instance discrimination for self-superivsed learning for SimCLR. Is this what is done? Should be described.

Naming of baselines: VGG refers to an architecture whereas ProtoNets refers to a training objective. That is, technically, one could use ProtoNet’s algorithm to train a VGG network. Because of this, this terminology is confusing. Instead of referring to the baseline as VGG, it should be referred to based on the algorithm used for training it (e.g. ‘finetuning’, as this is referred to in the few-shot learning community; if I understood correctly what VGG is in these experiments).

Correctness
===========
The authors mention that ‘We note that in the original ProtoNets paper (Snell et al., 2017), there is fine-tuning of weights after the initial pre-training.’ – this is incorrect, ProtoNet does not finetune weights at meta-testing. Its ‘learning’ happens in closed form that can be thought of as  learning a new gaussian classifier per task, but the weights of the network aren’t modified.

Related work
===========
There is no related work section in the paper! This is a very significant omission. As it stands, It is unclear how the proposed method differs from previous continual learning approaches that also incorporate replay (these are numerous; searching in my browser for the keywords ‘continual replay’ fills pages of results – e.g. ‘Experience Replay for Continual Learning’ by Rolnick et al, to name a specific one). Previous replay methods and their connection to the present experiments aren’t discussed in the paper.

Experiments
===========
Re: the performance is better in the deep configuration for the large classes tasks. The authors say that this is surprising since deep configuration implies seeing more support sets, and thus doing more rounds of finetuning, and thus more opportunities for catastrophic forgetting. However, there may be a simpler explanation: wider support sets (more classes presented within a given support set) need more steps of finetuning (which I understand here were capped for compute reasons), and spreading them out across more episodes implies more *total* steps of finetuning for those classes.

Re: ‘VGG accuracy increases as the instances are distributed over more support sets’ - the same might be true here.

‘CNNs may be more effective at continual learning with smaller support sets’ - to investigate this hypothesis properly, the total number of finetuning steps should be held constant between the two scenarios of wide vs deep.

Why isn’t replay added to ProtoNets? This seems straightforward: we would use the augmented support set (the one in the current episode together with the one sampled from the replay buffer) to compute the prototype of each class, and then proceed as usual. This would show whether replay can also help in the context of customizing only the classifier and not changing the features themselves. The ‘opposite’ direction can also be interesting to investigate: with the VGG model, one can just train a linear layer on top of the features for each new episode, while keeping the features frozen. If doing it this way, does replay still bring clear benefits as it does now? Running these experiments and adding some discussion about this would enrich the paper.

Motivation, Significance, Novelty
===========================
What is the connection between the two settings studied here (instance classification tasks and tasks with much larger ways)? It seems that they are orthogonal to each other and some motivation for why these two independent settings are studied here as the main contribution of the paper didn’t really come across.

The link to Hippocampal replay is not clear. The authors don’t really go into detail about what makes their replay approach (which seems very standard) be Hippocampal-like? The title of the paper makes it sound like this is central to the story of the paper, when this link is actually barely mentioned, and in fact empirically the proposed method with replay underperforms the simple baseline of ProtoNets almost always. It’s not really clear to me what is the take-away message from this paper. The authors hypothesize that replay can surpass the performance of ProtoNets if combined with more sophisticated models or tested on more challenging datasets, but this sounds like an empirical question that remains to be investigated.

Re: limitations: ‘Also, the models and tasks used are very computationally expensive, limiting
our ability to compare to some of the models used in the original work.’ - what does it mean for a task to be computationally expensive? It’s not clear what the difficulty is here. Expanding the experimental scope of the paper to include more methods and datasets would significantly increase the significance of the observations.

List of necessary changes
====================
Addressing the clarity/correctness concerns listed above is necessary for recommending acceptance. So is the addition of a comprehensive related work section, and enriching the experimental section. A plethora of models are proposed for few-shot and continual learning and comparing only 2 on only 1 dataset is far from meeting the bar of acceptance in my opinion.


**Strengths And Weaknesses:**

The paper studies an interesting and important problem relating to the intersection of continual and few-shot learning. The additional experimental settings explored here compared to previous work may provide a useful data point to the community, since it is believable that problems of interest may have many more than 20 classes, and the instance formulation also may have practical applicability. The paper is mostly well-written, but clarity is sometimes poor (see comments below).
The main drawback of the paper in my opinion is that the experimental analysis is very limited to a single dataset, Omniglot, which is a very simple one that the research community has largely moved on from when it comes to studying few-shot learning. Also, only two baselines are considered which makes it hard to really gain insights about the new setting that are studied, since this is an empirical investigation. Other downsides of the paper are the lack of clarity in key places and the lack of discussion of related work.

---

### Review · Reviewer_kZ9F · 2022-10-20

**Summary Of Contributions:**

- Paper develops a benchmark for continual few-shot learning
- Paper compares VGG, VGG+replay, ProtoNets to solving the task defined by the benchmark.

**Broader Impact Concerns:**

- The limitations section is very shallow and it does not analyze broader impacts and concerns. For example, one of the motivational lines in the intro is comparing current ML algorithms against capabilities of humans and animals. Again, title contains "Hippocampal-inspired replay". All that may create the impression that the authors are making a step towards closing the gap between AI and biological intelligence. Is this really the case in the article, is this the goal? What are gaps in the current frameworks that stall the progress in this direction? How can they be closed in the future? What are risks associated with AI acquiring human-like capabilities? It would be highly appreciated if authors could make a step back and address these higher-level questions in revision.

**Requested Changes:**

MAJOR POINTS

- "To our knowledge, none of the reported works explore continual learning with few samples per class." I know at least one more work in addition to (Antoniou et al., 2020) that explores continual learning with few samples per class. https://openaccess.thecvf.com/content_ICCV_2019/papers/Siam_AMP_Adaptive_Masked_Proxies_for_Few-Shot_Segmentation_ICCV_2019_paper.pdf. A more thorough review of literature combining few-shot and continual learning is required. Currently there is a lot of emphasis on continual learning and few-shot learning in separation. However, those are not primary focus areas of the paper. In contrast, Antoniou et al., 2020 has 36 citations currently. Simple google search "continual few-shot learning" provides a number of results. Please explore these venues and summarize the current state of continual few-shot learning domain in a comprehensive way, including benchmarks, tasks and architectural approaches.

- Once authors update the continual+few-shot learning part of the literature review, it would be advisable to iterate on the motivation part of the paper. Please clearly describe the research gap you are closing and the research questions you are targeting to answer in your study. So far, this part does not seem to be particularly convincing to me, because (i) literature review seems to be shallow and (ii) the research gap and research questions parts are not clearly identified.

- Introduction does not have a section clearly describing contributions. Please consider adding it.

- Source code is unavailable. Please release or provide a clear timeline when it will be available.

- Section 2.1 lacks details. For example, the significance of CCI is unclear, what does the class-change interval part of the framework model?

- Section 2.2, CFSL at scale jumps into technical details without much context and motivation. Why at scale? I don't understand the significance of "lowering from 250 epochs and 500 iterations per epoch to 10 epochs and 100 iterations per epoch". In my mind, this is the kind of detail that is best kept for empirical results section, or even for appendix. Same comments largely apply to Sections 2.2.1 and 2.2.2. Subsection 2.2.2 addresses the motivation part, but in a very superficial way. example, the deep and wide concepts are introduced, but without explaining their connections to practical application scenarios and examples that would help to appreciate the significance of scaling.

- Section 2.3 develops the instance level classification. Instance-level reasoning is quite prominent in instance segmentation. For example , this paper https://arxiv.org/abs/2105.05312 considers incremental Few-Shot Instance Segmentation. There is one-shot instance segmentation benchmark https://paperswithcode.com/task/one-shot-instance-segmentation. It would be highly appreciated if the authors could review literature in this area and discuss connections with their work.

- Section 3 does not have a detailed description of the empirical experiments setup. Please add the detailed description of datasets used, compute infrastructure, how much time it takes to reproduce the experiments, deep learning framework used, optimizers and hyperparameter settings employed.

- I do not understand the significance of Section 3.1.1. Is it that "ProtoNets are substantially more accurate than VGG"? Why is the section called replication then? Which column in Table 2 is taken from (Antoniou et al., 2020)? Please explain what conclusion can be drawn from presented results.

- Section 3.1.2 describes results presented in Table 3 and Figure 4. It does not draw any conclusions or extracts any insights from the data. Why was this experiment set executed, what does it demonstrate? Hence, I have an impression that the section can be simply deleted from the text, unless authors provide non-trivial insights and conclusions based on this experiment set.

- Section 3.2, similar concerns as subsections 3.1.1 and 3.1.2. VGG+replay is better than VGG, but worse than ProtoNets. What does it mean, we should use ProtoNets to solve this task? What about ProtoNets+replay, will it be even better than all other techniques? Same high level problem: experiment design is not motivated, research question is unclear. Hence the significance of results is very hard to appreciate.

- I do not think that having a separate discussion section devoted to detailed analysis of experiments is a good idea at all. The experiment design, motivation and detailed analysis of results should happen before and during reporting the experiments. Otherwise reader has no chances of following the key ideas behind the experiments and appreciating the results. In my view, the discussion section should focus on high-level generalizations that can be extracted from individual experiments and provide forward looking perspective on the conclusions drawn from current research, limitations of the research and ways of addressing these limitations through extensions of the current research. I propose that the content of section 4 be used to better motivate and explain the significance of individual experiments in section 3, while section 4 be rewritten to analyze high-level generalizations, limitations of current research and ways forward to address them.

- I am not sure that the experiment design is valid in terms of comparing ProtoNets and VGG. It is unclear from the experiment design, which architecture is used in ProtoNets. If this is 4 x Conv64 from the original paper and VGG is VGG16 with 16 layer, I do not really understand how those can be compared on equal foot. The underlying backbone in ProtoNets and VGG approaches should be the same to make the comparison sound. I am not at all sure this is the case.

- Since a lot of the focus of the paper is on defining the new continual few-shot learning benchmark, I would imagine spending a lot more time in the main body and in appendices on reporting the exploratory data analysis results of the dataset and respective task would be important. I do not see a lot of this in the paper.

- The title is "Continual few-shot learning with Hippocampal-inspired replay", hinting the the key contribution is replay for VGG? Looking at the tables, it appears to me that the replay approach with VGG is not beating ProtoNets. What is the point then? Having read the paper, I would think a complete shift in focus of the paper is needed towards making it a dataset paper. Something like "Continual few-shot learning benchmark", where 'benchmark' is filled with more details, perhaps about the actual contribution related to the deep and wide aspects?

MINOR POINTS

- Ideally, Figure 1 appears on the same page as Section 2.1 describing the frameowork
- "Task D (see Figure 1) introduces both new classes and multiple instances of each class, which we argue is the most common, applicable real-world scenario." First, could you please rephrase and make it more clear, I don't think I 100% understand what you mean here, especially "both new classes and multiple instances of each class". Second, could you please provide a few real-world examples to make sure that your claim is supported?




**Strengths And Weaknesses:**

STRENGTHS

- Interesting and relevant topic
- Empirical results are presented with confidence intervals

WEAKNESSES

- Literature review incomplete
- Research gap and research questions are not clearly identified, motivation is weak
- Clarity must be improved, see specific points in Requested Changes section
- Contributions are not clearly stated
- Limitations section is shallow and analysis of broader impacts is missing
- Paper has no flow, things are presented in isolated fashion without transitions and overall story behind the narrative
- Exploratory data analysis results of the dataset and benchmark are not reported
- Experimental design is potentially flawed
   - architecture complexity VGG 16 layer vs ProtoNets 4 x Conv64
   - ProtoNets can be tested with replay, further boosting their performance.

---

### Note · Authors · 2022-11-13

**Comment:**

Thanks to the editor and reviewer for feedback.
We are working on a revised manuscript with additional experiments.
Given the timeframe, we are withdrawing and intend to resubmit.

**Withdrawal Confirmation:**

I have read and agree with the venue's withdrawal policy on behalf of myself and my co-authors.

---

> ### Author Response · Authors · 2023-11-07
> **Re-submission**
>
> We addressed the feedback and re-submitted it in July 2023, here: https://openreview.net/forum?id=aPyXAqeg3r&noteId=ta9NUSTsj0